# Association between the composite dietary antioxidant index and constipation: Evidence from NHANES 2005–2010

Shouxin Wei[1]*, Sijia Yu[2], Yunsheng Lan[1], Yingdong Jia[1]

1 Department of Gastrointestinal Surgery, Suining Central Hospital, Suining, Sichuan Province, China,
2 Department of General Practice, Suining Central Hospital, Suining, Sichuan Province, China

* 1079656665@qq.com

**Data Availability Statement:** The datasets generated and/or analyzed during the current study are available in the NHANES repository, [https://www.cdc.gov/nchs/nhanes/].

## Abstract

### Background

Dietary antioxidants have been found to improve various diseases, including digestive, cardiovascular, and urinary disorders. However, the relationship between CDAI and constipation remains unclear. This study aims to investigate the potential link between CDAI and constipation among adults in the United States.

### Methods

This study utilized data from the National Health and Nutrition Examination Survey (NHANES) collected between 2005 and 2010. Dietary data from the respondents were obtained through two 24-hour dietary recall interviews. Multiple statistical methods, including multivariable logistic regression, subgroup analysis, and smooth curve fitting analysis, were used to explore the association between CDAI and chronic constipation.

### Results

A total of 10,904 participants were included in the study, of whom 1,184 were identified as having chronic constipation. After adjusting for potential confounders, multivariable logistic regression analysis showed that higher CDAI was significantly associated with a lower risk of constipation (OR = 0.958 [0.929, 0.987]). Compared to the lowest quartile, the highest quartile of CDAI was associated with a significantly reduced prevalence of constipation (OR = 0.704 [0.535, 0.927]). Subgroup analysis indicated that differences in gender, alcohol intake, and smoking status might influence the association between CDAI and constipation. Smooth curve analysis revealed an "n" shaped relationship between CDAI and constipation among non-alcohol consumers, with a turning point at a CDAI value of 1.08.

### Conclusion

An elevated CDAI is negatively correlated with the incidence of chronic constipation, suggesting that increasing dietary antioxidant intake may reduce constipation prevalence.

**Funding:** The author(s) received no specific funding for this work.

**Competing interests:** The authors have declared that no competing interests exist.

These findings underscore the importance of dietary antioxidants in maintaining gut health and provide comprehensive guidance for clinical and public health practices.

## 1 Introduction

Constipation is a common gastrointestinal disorder characterized by infrequent bowel movements, hard stools, and difficulty in defecation. Its global prevalence ranges from 2.6% to 26.9%, with higher rates observed among women and the elderly [1]. Constipation significantly reduces patients' quality of life and imposes a substantial disease burden. Chronic constipation may lead to severe complications such as gastrointestinal obstruction, perforation, and bowel necrosis, potentially threatening patients' lives [2, 3].

Constipation is influenced by various factors, including age, medication use, lifestyle, and stress, with dietary factors being a significant contributor [4]. Research indicates that dietary fiber can increase stool volume, promote bowel movements, and improve stool frequency and consistency [5]. Conversely, a low-fiber diet can decrease stool bulk and prolong colonic transit time, leading to constipation. Adequate water intake can soften stools and reduce colonic transit time, alleviating constipation symptoms [6]. Recently, the role of dietary antioxidants in constipation has garnered attention. Antioxidants in the diet, such as vitamins C and E and polyphenols, may promote gut health and function by reducing oxidative stress and inflammation in the intestines [7]. They might also balance gut microbiota, increasing beneficial bacteria and improving the intestinal environment, thus reducing constipation risk [8].

TheCDAI is a quantitative measure of the overall intake of dietary antioxidants. It incorporates the intake of vitamins A, C, and E, zinc, selenium, and carotenoids, generating an index score through weighted calculations. Unlike single antioxidant intake measurements, the CDAI provides a more comprehensive reflection of an individual's dietary antioxidant capacity. Researchers have increasingly focused on the potential applications of CDAI in health and disease prevention. Huiqin et al. [9] explored the association between CDAI and aging, finding that each standard deviation increase in CDAI was associated with a 0.18-year reduction in phenotypic age. Min et al. [10] analyzed NHANES data from 2011–2018 and found a negative correlation between CDAI and the prevalence of chronic kidney disease in American adults, suggesting that high antioxidant intake might reduce the risk of chronic kidney disease. However, no studies have yet examined the relationship between CDAI and constipation.

This study used data from three consecutive cycles (2005–2010) of NHANESto analyze various demographic, lifestyle, and dietary factors. The goal was to explore the potential link between CDAI and constipation in a representative sample of the U.S. population. The findings will provide scientific evidence for understanding the impact of dietary antioxidant intake on constipation prevention and management and may inform public health policy development.

## 2 Methods

NHANES, led by the Centers for Disease Control and Prevention (CDC), aims to provide information on the health and nutritional status of the U.S. population. By conducting face-to-face interviews and physical examinations with thousands of Americans, NHANES collects health and nutrition data to monitor and assess health risks and disease burden. This information supports public health policies and interventions by government, healthcare institutions, and researchers.

Data for this study were drawn from the 2005–2006, 2007–2008, and 2009–2010 NHANES cycles, including 31,034 participants. Exclusion criteria were: (1) age < 20 years (n = 13,902),

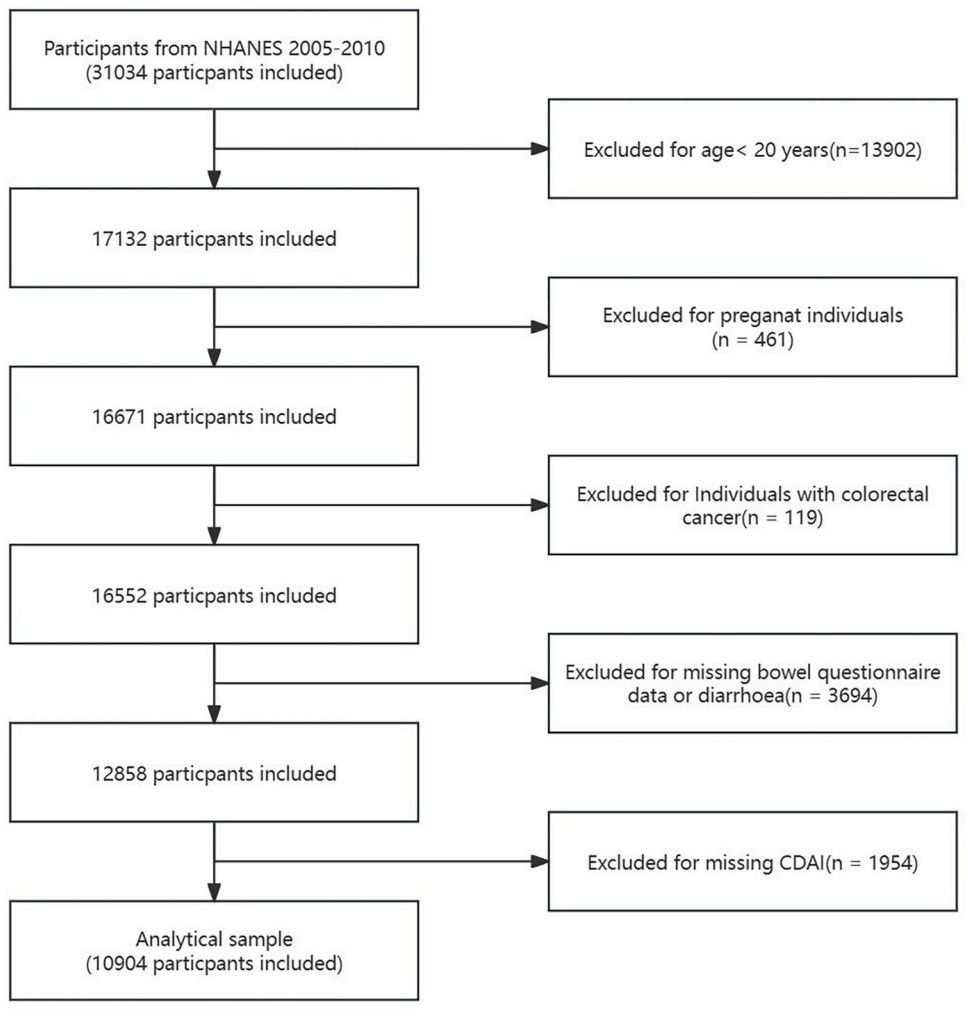

**Fig 1. Flowchart of the study.**

(2) pregnancy (n = 461), (3) self-reported colorectal cancer (n = 119), (4) missing bowel health questionnaire data or self-reported chronic diarrhea (n = 3,694), and (5) missing dietary antioxidant data needed to calculate CDAI (n = 1,954). Ultimately, 10,904 participants were included in the final analysis, comprising 5,378 men and 5,526 women (Fig 1).

## 2.1 Definition of CDAI

The Comprehensive Dietary Antioxidant Index (CDAI) is a method to evaluate the total amount of antioxidants in the diet. Based on previous research, we calculated the CDAI values, which include antioxidants such as vitamins A, C, and E, zinc, selenium, and carotenoids [11]. The calculation method involves subtracting the mean intake from the individual intake for each antioxidant, dividing by the standard deviation to standardize the values, and then summing these standardized values to obtain the total CDAI. The formula is as follows:

$$CDAI = \sum_{i=1}^{n=6} \frac{x_i - \mu_i}{S_i}$$

$x_i$ indicates dietary antioxidant daily intake; $\mu_i$ indicates mean dietary antioxidant daily intake; $S_i$ indicates standard deviation.

## 2.2 Definition of constipation

Between 2005 and 2010, NHANES assessed constipation in participants through the Bowel Health Questionnaire (BHQ) conducted in Mobile Examination Centers (MEC). The Bristol Stool Form Scale (BSFS) and a bowel movement frequency questionnaire were used to evaluate constipation. Participants were shown a card depicting different stool types and asked to identify the number corresponding to their usual stool type. Types 1 (separate hard lumps) and 2 (sausage-shaped but lumpy) were defined as constipation. Types 3 (sausage-shaped with cracks on the surface), 4 (like a sausage or snake, smooth and soft), and 5 (soft blobs with clear-cut edges) were considered normal. Types 6 (fluffy pieces with ragged edges, mushy) and 7 (watery, no solid pieces) were defined as diarrhea. Participants also reported their bowel movement frequency: "How many times do you usually have a bowel movement per week?" Those with fewer than 2 bowel movements per week were defined as constipated, those with 3 to 21 movements per week were normal, and those with more than 22 movements per week were classified as having diarrhea.

## 2.3 Covariates

To minimize potential confounding factors, we reviewed relevant literature and collected covariates related to constipation and CDAI, including age, gender, race, education level, marital status, poverty income ratio (PIR), BMI, depression status, vigorous physical activity, alcohol intake, smoking status, hypertension, diabetes, lung disease, heart disease, liver disease, protein intake, carbohydrate intake, fiber intake, fat intake, water intake, and energy intake. Race was categorized into Mexican American, non-Hispanic black, non-Hispanic white, other Hispanic, and other race. Marital status was classified as married/cohabiting, divorced/separated/widowed, and never married. Educational attainment was divided into less than high school, high school, and more than high school. Poverty income scores were grouped into $< 2$ and $\geq 2$. BMI classifications were underweight/normal ($< 25.0$ kg/m$^2$), overweight (25.0–29.9 kg/m$^2$), and obese ($>29.9$ kg/m$^2$). Assessed using the PHQ-9 questionnaire, with a score of 10 or higher considered positive for depression. Definitions of vigorous exercise varied by NHANES year. In 2005–2006, it included activities resulting in profuse sweating and significantly increased respiration or heart rate for at least 10 minutes in the past 30 days. In 2007–2010, it was defined as work involving strenuous activity that markedly increased respiration or heart rate for at least 10 consecutive minutes, such as lifting heavy objects or construction work. Alcohol consumption was defined as having 12 or more drinks in a year. Smoking status was categorized into former smokers (those who had smoked more than 100 cigarettes but were not currently smoking), current smokers (those who were still smoking and had smoked at least 100 cigarettes in their lifetime), and non-smokers (those who had never smoked or had smoked less than 100 cigarettes). Hypertension was defined by (1) a physician's diagnosis, (2) average systolic blood pressure $\geq 130$ mmHg or diastolic blood pressure $\geq 80$ mmHg, (3) use of antihypertensive medication. Similarly, diabetes was defined by (1) a physician's diagnosis, (2) HbA1c level $\geq 6.5\%$, (3) use of diabetes medication or insulin.

## 2.4 Statistical analysis

Continuous variables were expressed as mean ± standard deviation (SD) and compared using t-tests. Categorical variables were expressed as percentages and compared using chi-square tests. To analyze the relationship between CDAI and constipation, we performed multivariable

logistic regression. The crude model made no adjustments. Model 1 adjusted for age, gender, and race. Model 2 adjusted for age, gender, race, education level, marital status, PIR, BMI, depression status, vigorous physical activity, alcohol intake, smoking status, hypertension, diabetes, lung disease, heart disease, liver disease, protein intake, carbohydrate intake, fiber intake, fat intake, water intake, and energy intake. Results were expressed as odds ratios (OR) with 95% confidence intervals (95% CI). A p-value < 0.05 was considered statistically significant. All analyses were conducted using R (version 4.3.2, http://www.R-project.org) and Empower-Stats software (version 2.0, http://www.empowerstats.com).

## 3 Results

### 3.1 Baseline characteristics

The characteristics of the study participants are presented in Table 1. This study included 10,904 participants, of whom 1,184 were constipated and 9,720 were healthy. The mean age was 48.07±18.30 years, and 49.32% were male. The average CDAI score was 0.05 ± 3.80. Significant differences were found between the constipation and normal groups in all baseline variables except for vigorous physical activity, diabetes, heart disease, and liver disease (p < 0.05). Constipated individuals were more likely to be female, non-Hispanic Black, less educated, divorced/separated/widowed, never married, have a lower PIR, underweight/normal weight, depressed, non-drinkers, non-smokers, without hypertension, and have lung disease. Compared to the lowest CDAI quartile, the highest quartile was characterized by a higher proportion of males, non-Hispanic Whites, higher education levels, married or cohabiting status, higher PIR, underweight/normal weight, absence of depression, vigorous physical activity, alcohol consumption, former or never smokers, absence of hypertension, diabetes, lung disease, heart disease, and non-constipation. Constipated patients had significantly lower CDAI scores than non-constipated individuals (p < 0.05).

### 3.2 Distribution and concentration of CDAI in constipated patients

The distribution and concentration of each element of the CDAI in constipated patients are presented in Table 2. The median CDAI was -0.54 (-0.63, -0.45). The daily intakes for specific nutrients were as follows: vitamins A, C, and E were 515.00 (502.45, 527.55) μg/day, 67.08 (65.34, 68.82) mg/day, and 6.23 (6.09, 6.36) mg/day, respectively. Zinc intake was 10.22 (10.05, 10.38) mg/day, selenium was 99.23 (98.15, 100.31) μg/day, and carotenoids were 6570.75 (6358.77, 6782.73) μg/day.

### 3.3 Association between CDAI and constipation

Table 3 shows the results of the multivariate linear analysis between CDAI and constipation. CDAI scores were analyzed both as continuous variables and as quartiles, with multiple models adjusted for different covariates. The results indicate a significant negative correlation between CDAI scores and constipation when CDAI was considered as a continuous variable (Fig 2). This suggests that higher CDAI scores are associated with a lower risk of constipation. In Model 2, this relationship remained significant (OR = 0.958 [0.929, 0.987]). Quartile analysis demonstrated a dose-response relationship between CDAI scores and the risk of constipation. Higher CDAI scores were associated with a lower risk of constipation. In Model 2, compared to the lowest quartile, the highest CDAI quartile showed a more significant negative correlation with constipation (OR = 0.704 [0.535, 0.927], ptrend < 0.05).

**Table 1. Baseline characteristics of the general adult population divided by constipation status in the NHANES 2005–2010.**

| Characteristics | Total (n = 10904) | No constipation(n = 9720) | constipation(n = 1184) | P-value |
|---|---|---|---|---|
| Age (years) | 50.28±17.80 | 50.55±17.72 | 48.07±18.30 | <0.001 |
| Sex, n (%) | | | | <0.001 |
| Male | 5378 (49.32%) | 5050 (51.96%) | 328 (27.70%) | |
| Female | 5526 (50.68%) | 4670 (48.05%) | 856 (72.30%) | |
| Race, n (%) | | | | <0.001 |
| Mexican American | 1851(16.98%) | 1669 (17.17%) | 182 (15.37%) | |
| Other Hispanic | 902(8.27%) | 783 (8.06%) | 119 (10.05%) | |
| Non-Hispanic White | 5643(51.75%) | 5107 (52.54%) | 536 (45.27%) | |
| Non-Hispanic Black | 2107(19.32%) | 1800 (18.52%) | 307 (25.93%) | |
| Other Race—Including Multi-Racial | 401(3.68%) | 361 (3.71%) | 40 (3.38%) | |
| Education level, n (%) | | | | <0.001 |
| Below high school | 2794(25.62%) | 2424 (24.94%) | 370 (31.25%) | |
| High school | 2611(23.95%) | 2285 (23.51%) | 326 (27.53%) | |
| Above high school | 5499(50.43%) | 5011 (51.55%) | 488 (41.22%) | |
| Marital status, n (%) | | | | <0.001 |
| Married or living with partner | 6782(62.20%) | 6123 (62.99%) | 659 (55.66%) | |
| Divorced, separated, or widowed | 2415(22.15%) | 2113 (21.74%) | 302 (25.51%) | |
| Never married | 1707(15.65%) | 1484 (15.27%) | 223 (18.83%) | |
| Family PIR, n (%) | | | | <0.001 |
| < 2 | 4425(40.58%) | 3841 (39.52%) | 584 (49.32%) | |
| ≥ 2 | 6479(59.42%) | 5879 (60.48%) | 600 (50.68%) | |
| BMI, n (%) | | | | <0.001 |
| < 25.0kg/m2 | 3108(28.50%) | 2697 (27.75%) | 411 (34.71%) | |
| 25.0–29.9 kg/m2 | 3744(34.34%) | 3363 (34.60%) | 381 (32.18%) | |
| ≥29.9kg/m2 | 4052(37.16%) | 3660 (37.65%) | 392 (33.11%) | |
| PHQ Score | | | | <0.001 |
| < 10 | 880(8.07%) | 700 (7.20%) | 180 (15.20%) | |
| ≥ 10 | 10024(91.93%) | 9020 (92.80%) | 1004 (84.80%) | |
| Vigorous physical activity, n (%) | | | | 0.153 |
| Yes | 1453(13.33%) | 1311 (13.49%) | 142 (11.99%) | |
| No | 9451(86.67%) | 8409 (86.51%) | 1042 (88.01%) | |
| Drinking status, n (%) | | | | <0.001 |
| Yes | 7838(71.88%) | 7102 (73.067%) | 736 (62.16%) | |
| No | 3066(28.12%) | 2618 (26.93%) | 448 (37.84%) | |
| Smoking status, n (%) | | | | <0.001 |
| Former | 2848(26.12%) | 2608 (26.83%) | 240 (20.27%) | |
| Current | 2259(20.72%) | 2008 (20.66%) | 251 (21.20%) | |
| Never | 5797(53.16%) | 5104 (52.51%) | 693 (58.53%) | |
| Hypertension, n (%) | | | | <0.001 |
| Yes | 5493(50.38%) | 4968 (51.11%) | 525 (44.34%) | |
| No | 5411(49.62%) | 4752 (48.89%) | 659 (55.66%) | |
| Diabetes, n (%) | | | | 0.346 |
| Yes | 1601(14.68%) | 1438 (14.79%) | 163 (13.77%) | |
| No | 9303(85.32%) | 8282 (85.21%) | 1021 (86.23%) | |
| Pulmonary disease, n (%) | | | | 0.009 |
| Yes | 1681(15.42%) | 1468 (15.10%) | 213 (17.99%) | |
| No | 9223(84.58%) | 8252 (84.90%) | 971 (82.01%) | |

(*Continued*)

**Table 1.** (Continued)

| Characteristics | Total (n = 10904) | No constipation(n = 9720) | constipation(n = 1184) | P-value |
|---|---|---|---|---|
| Heart disease, n (%) | | | | 0.352 |
| Yes | 944(8.66%) | 833 (8.57%) | 111 (9.38%) | |
| No | 9960(91.34%) | 8887 (91.43%) | 1073 (90.63%) | |
| Liver disease, n (%) | | | | 0.165 |
| Yes | 370(3.39%) | 338 (3.48%) | 32 (2.70%) | |
| No | 10534(96.61%) | 9382 (96.52%) | 1152 (97.30%) | |
| Protein intake, n (%) | 78.40±31.32 | 79.56±31.52 | 69.16±28.00 | <0.001 |
| Carbohydrate intake, n (%) | 244.66±93.97 | 246.16±94.31 | 232.29±90.24 | <0.001 |
| Dietary fiber intake, n (%) | 16.07±8.02 | 16.33±8.10 | 13.96±7.05 | <0.001 |
| Total fat intake, n (%) | 74.20±33.34 | 75.20±33.57 | 66.29± 30.20 | <0.001 |
| Moisture intake, n (%) | 2714.71±1159.15 | 2754.67±1163.58 | 2386.66±1067.36 | <0.001 |
| Energy intake, n (%) | 1988.61±714.91 | 2010.42± 718.58 | 1809.54±657.47 | <0.001 |
| CDAI | 0.05 ± 3.80 | 0.19 ± 3.84 | -1.06 ± 3.24 | <0.001 |

PIR: Poverty Income Ratio; BMI: Body Mass Index; CDAI: Composite Dietary Antioxidant Index; n: Number of study samples.

## 3.4 Association between the components of CDAI and constipation

We further analyzed the relationship between individual components of CDAI and constipation(Fig 2; Table 4). In the quartile analysis, compared to the lowest quartile, individuals in the highest quartile of Vitamin A intake had a 23.1% lower likelihood of experiencing constipation (OR = 0.769 [0.621, 0.951], ptrend = 0.022). Additionally, those in the highest quartile of carotenoid intake had a 19.8% lower likelihood of experiencing constipation (OR = 0.802 [0.656, 0.979], ptrend = 0.193).

## 3.5 Subgroup analysis

We conducted subgroup analyses based on variables such as age, gender, race, alcohol intake, smoking status, hypertension, and diabetes. As shown in Fig 3, the results indicated significant differences in the association between CDAI scores and constipation among subgroups of gender, alcohol intake, and smoking status (P for interaction < 0.05). No significant interaction effects were observed in other subgroups (P for interaction > 0.05). Further smooth curve fitting analysis was performed, stratified by alcohol intake. The results showed a non-linear relationship between CDAI and constipation in both the drinking group (edf = 1.40, p = 0.001) and the non-drinking group (edf = 2.93, p = 0.33), with inflection points at 3.79 and 1.08, respectively (Fig 4 and Table 5).

**Table 2. Distribution and concentration of CDAI and its elements in adults with constipation in the NHANES 2005–2010.**

| | Mean | 5th | 25th | 50th | 75th |
|---|---|---|---|---|---|
| CDAI | 0.05 | -4.80 | −2.56 | -0.54 | 1.89 |
| Vitamins A, µg/day | 618.10 | 142.08 | 322.88 | 515.00 | 781.00 |
| Vitamins C, mg/day | 86.08 | 10.05 | 32.99 | 67.08 | 117.68 |
| Vitamins E, mg/day | 7.57 | 2.43 | 4.28 | 6.23 | 8.95 |
| Zinc, mg/day | 11.47 | 4.65 | 7.45 | 10.21 | 13.91 |
| Selenium, µg/day | 106.24 | 45.77 | 73.75 | 99.23 | 130.15 |
| Carotenoid, µg/day | 9208.70 | 827.08 | 3188.38 | 6570.75 | 12255.75 |

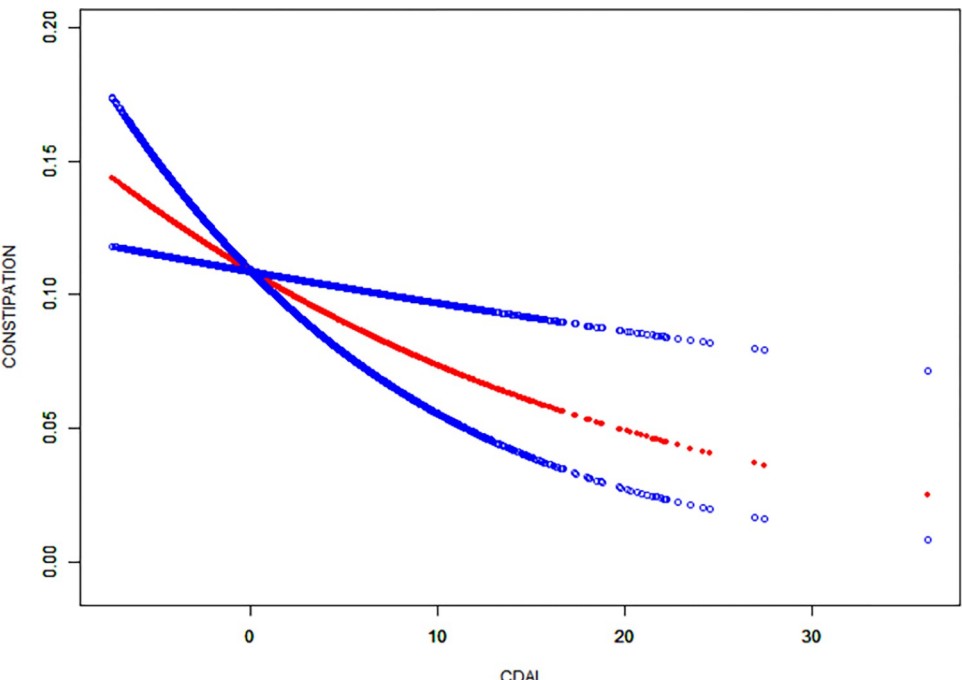

**Fig 2. Exposure-response relationship between CDAI scores and constipation in NHANES from 2005–2010.** The model is adjusted for gender, age, race, education level, marital status, PIR, BMI, depression, vigorous physical activity, alcohol consumption, smoking, hypertension, diabetes, lung disease, heart disease, liver disease, protein intake, carbohydrate intake, fiber intake, fat intake, water intake, and energy intake.

## 4 Discussion

Chronic constipation is a significant global public health issue that severely impacts patients' quality of life and overall health. Dietary adjustments are fundamental in the prevention and treatment of chronic constipation. Exploring the association between antioxidant-rich diets and constipation may address this widespread health problem. CDAIis a novel index that quantifies and evaluates the overall antioxidant capacity of a diet by measuring the intake of multiple antioxidants. Previous studies have found that higher CDAI scores are associated with reduced incidences of stroke, kidney stones, and depression [12, 13]. In this NHANES

**Table 3. Odds ratios and 95% confidence intervals for CDAI scores and constipation in the NHANES 2005–2010.**

| Exposure | Crude OR (95% CI) | Model 1 OR (95% CI) | Model 2 OR (95% CI) |
|---|---|---|---|
| Continuous CDAI | 0.899 (0.881, 0.916) <0.00001 | 0.929 (0.910, 0.947) <0.00001 | 0.958 (0.929, 0.987) 0.00528 |
| Quartiles of CDAI | | | |
| Quartile 1 | 1[Reference] | 1[Reference] | 1[Reference] |
| Quartile 2 | 0.680 (0.581, 0.796) <0.00001 | 0.741 (0.631, 0.869) 0.00024 | 0.849 (0.711, 1.013) 0.06942 |
| Quartile 3 | 0.606 (0.515, 0.712) <0.00001 | 0.733 (0.621, 0.865) 0.00024 | 0.930 (0.754, 1.148) 0.49878 |
| Quartile 4 | 0.377 (0.314, 0.453) <0.00001 | 0.500 (0.414, 0.605) <0.00001 | 0.704 (0.535, 0.927) 0.01229 |
| P for trend | <0.00001 | <0.00001 | 0.026225 |

OR: Odds Ratio, 95% CI: 95% Confidence Interval. Crude model: Unadjusted for covariates. Model 1: Adjusted for gender, age, and race. Model 2: Adjusted for gender, age, race, education level, marital status, PIR, BMI, depression, vigorous exercise, alcohol consumption, smoking, hypertension, diabetes, lung disease, heart disease, liver disease, protein intake, carbohydrate intake, fiber intake, fat intake, water intake, and energy intake.

**Table 4. Odds ratios and 95% confidence intervals for each element of the CDAI and constipation in the NHANES 2005–2010.**

| Exposure | Vitamins A OR (95% CI) P | Vitamins C OR (95% CI) P | Vitamins E OR (95% CI) P | Zinc OR (95% CI) P | Selenium OR (95% CI) P | Carotenoid OR (95% CI) P |
|---|---|---|---|---|---|---|
| Quartile 1 | 1[Reference] | 1[Reference] | 1[Reference] | 1[Reference] | 1[Reference] | 1[Reference] |
| Quartile 2 | 0.915 (0.772, 1.084) | 0.816 (0.683, 0.975) | 0.967 (0.812, 1.152) | 0.985 (0.826, 1.176) | 0.931 (0.779, 1.113) | 0.844 (0.711, 1.002) |
| Quartile 3 | 0.934 (0.778, 1.123) | 0.909 (0.759, 1.089) | 0.907 (0.739, 1.114) | 0.943 (0.762, 1.166) | 0.832 (0.665, 1.042) | 1.041 (0.873, 1.240) |
| Quartile 4 | 0.769 (0.621, 0.951) | 0.882 (0.725, 1.073) | 0.928 (0.733, 1.175) | 1.174 (0.896, 1.539) | 0.757 (0.550, 1.041) | 0.802 (0.656, 0.979) |
| P for trend | 0.022 | 0.511 | 0.444 | 0.483 | 0.066 | 0.193 |

OR: Odds Ratio, 95% CI: 95% Confidence Interval. The model is adjusted for gender, age, race, education level, marital status, PIR, BMI, depression, vigorous exercise, alcohol consumption, smoking, hypertension, diabetes, lung disease, heart disease, liver disease, protein intake, carbohydrate intake, fiber intake, fat intake, water intake, and energy intake.

cross-sectional study, we included 10,904 participants, with an overall constipation prevalence of approximately 10.86%. Among these participants, 27.70% were male and 72.30% were female, consistent with previous findings that constipation prevalence is significantly higher in women [14]. The results indicated a negative correlation between CDAI and constipation after adjusting for all confounders (OR = 0.958 [0.929, 0.987]). The reduction in constipation prevalence was particularly significant in the highest CDAI quartile compared to the lowest quartile (OR = 0.704 [0.535, 0.927], ptrend < 0.05). This negative correlation between CDAI and

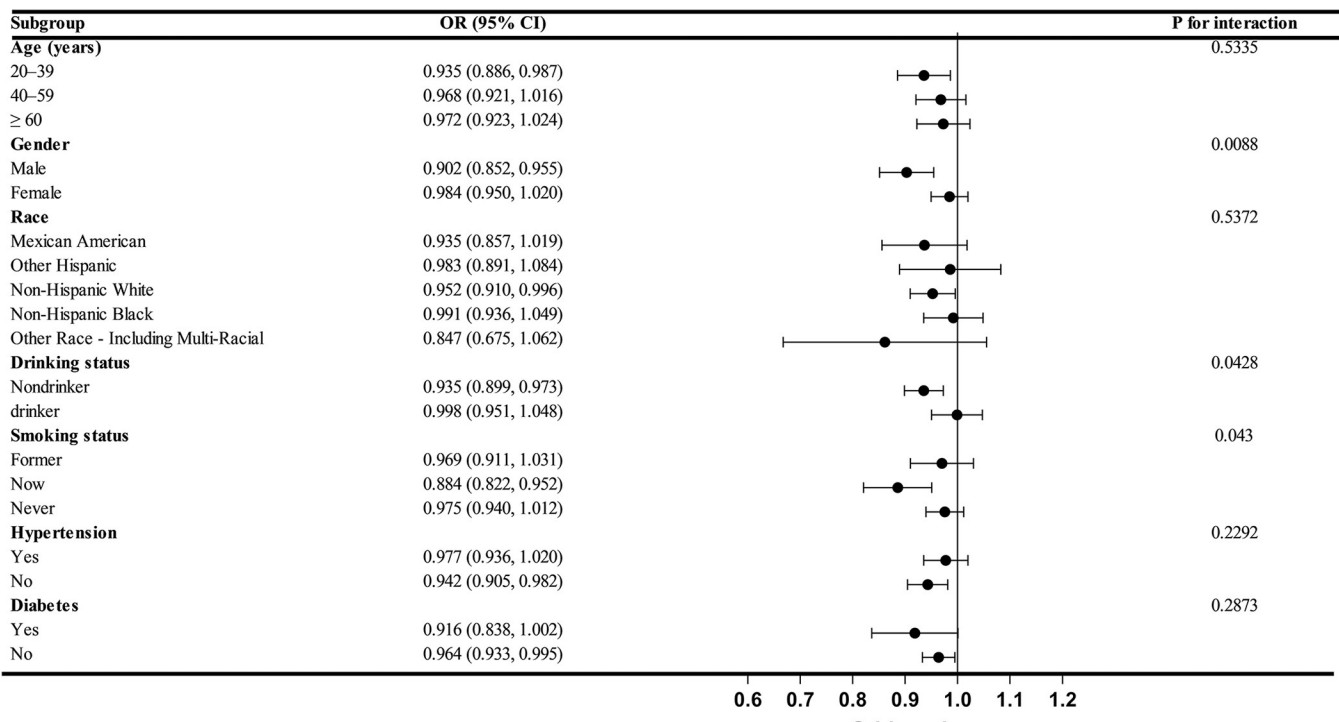

**Fig 3. Association between CDAI and constipation in different subgroups in NHANES from 2005–2010.** Adjustments were made for gender, age, race, education level, marital status, PIR, BMI, depression, vigorous physical activity, alcohol consumption, smoking, hypertension, diabetes, lung disease, heart disease, liver disease, protein intake, carbohydrate intake, fiber intake, fat intake, water intake, and energy intake.

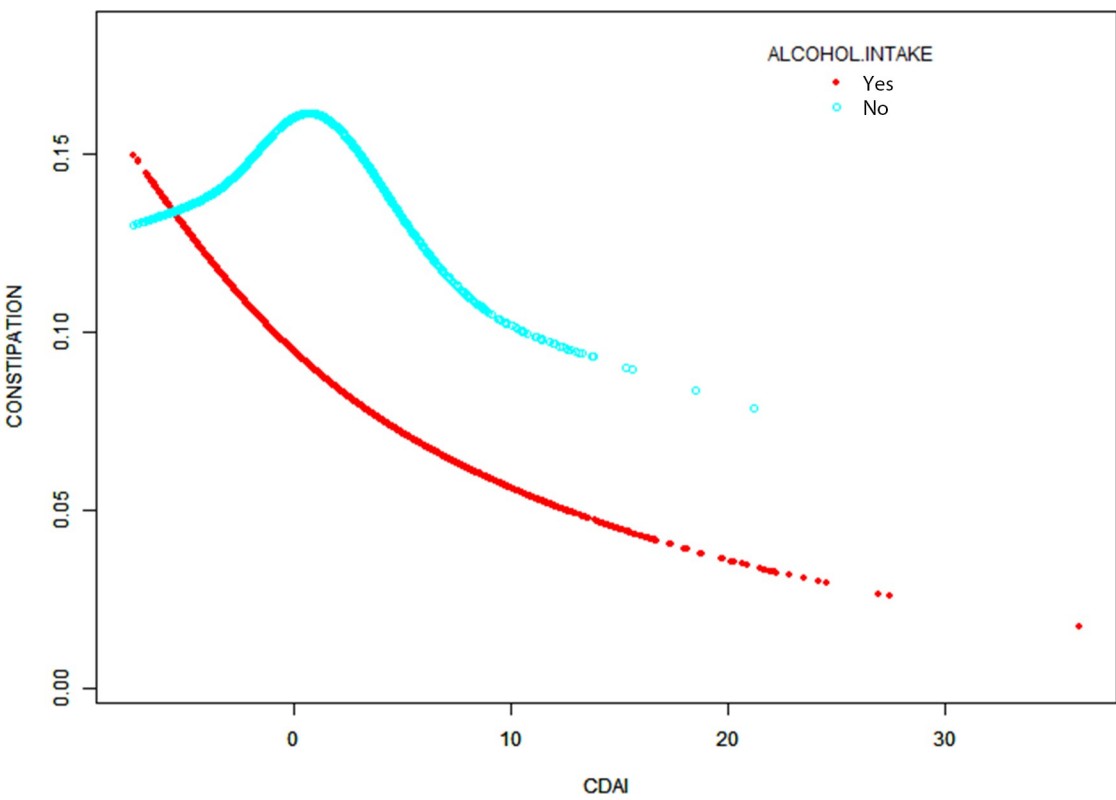

**Fig 4. Exposure-response relationship between CDAI scores and constipation stratified by alcohol consumption status in NHANES from 2005–2010.** The model is adjusted for gender, age, race, education level, marital status, PIR, BMI, depression, vigorous physical activity, alcohol consumption, smoking, hypertension, diabetes, lung disease, heart disease, liver disease, protein intake, carbohydrate intake, fiber intake, fat intake, water intake, and energy intake.

constipation was more pronounced in younger males, possibly due to more efficient nutrient absorption and utilization in younger individuals. As individuals age, the gastrointestinal neuromuscular system gradually deteriorates, leading to weakened peristaltic function in the elderly. Consequently, even with the support of a high CDAI diet, the protective effect against constipation is less pronounced in older adults compared to younger individuals [15, 16]. Similar patterns were observed in subgroups of individuals who consumed alcohol and those who smoked. Smoking and alcohol consumption can damage the intestinal mucosa, impairing absorption and motility functions. Antioxidants can protect the intestinal mucosa, promote

**Table 5. Threshold effect analysis of CDAI on constipation in non-drinking populations based on a two-segment linear regression model.**

| Constipation | Adjusted β (95% CI) P value |
| --- | --- |
| Alcohol intake: No | |
| Fitting by the standard linear model | 1.01 (0.96, 1.06 |
| Inflection point | 1.08 |
| CDAI < 1.08 | 1.08(1.00, 1.16) |
| | 0.0463 |
| CDAI > 1.08 | 0.93 (0.86, 1.01) |
| | 0.0966 |
| Log likelihood ratio | 0.012 |

cellular repair, and maintain normal intestinal function [17]. Smoking and drinking increase oxidative stress in the body, producing a large number of free radicals that damage cells and tissues. The antioxidants in a high CDAI diet can neutralize these free radicals, mitigating the negative effects of oxidative stress [18]. Therefore, the negative correlation between CDAI and constipation is more pronounced among smokers and drinkers. Additionally, in the non-drinking group, there is a complex nonlinear relationship between the CDAI and constipation. When the CDAI exceeds 1.08, there is a significant negative correlation between CDAI levels and the incidence of constipation. Conversely, when the CDAI is below 1.08, the relationship between CDAI levels and constipation incidence is positive. This might be because alcohol can enhance the bioavailability of antioxidants, allowing them to act more quickly [19]. In the non-drinking group, low levels of CDAI have a lesser impact on the gut, leading to an unstable effect. As the CDAI increases, the protective effects of antioxidants gradually become evident. To further understand the relationship between CDAI and constipation in non-drinking populations, more large-scale prospective studies are needed.

Dietary antioxidants, including vitamins, carotenoids, and minerals such as zinc and selenium, play a crucial role in maintaining human health and preventing diseases. They neutralize free radicals and protect cells from oxidative damage, thereby reducing the risk of diseases related to oxidative stress [20]. Numerous studies have shown that dietary antioxidants are associated with a reduced incidence of cardiovascular, urinary, endocrine diseases, and certain types of cancer [21]. Additionally, antioxidants help enhance immune function, improve skin health, slow down the aging process, and prevent neurodegenerative diseases [22].

Vitamins, carotenoids, and selenium act as cofactors for antioxidant enzymes, working together with superoxide dismutase and glutathione peroxidase to eliminate free radicals and prevent them from transforming into intracellular molecules, thereby reducing oxidative stress [23]. Vitamin A is crucial for maintaining normal vision, immune function, and skin health, but few studies have examined its relationship with constipation. Our study shows a negative correlation between high levels of vitamin A and constipation, possibly because vitamin A helps maintain the integrity of the intestinal mechanical barrier and promotes healthy and functional intestinal epithelial cells, enhancing motility and normal defecation. The intestinal mechanical barrier, composed of tight junction proteins such as ZO-1, occludin, and claudin, regulates intestinal permeability. Vitamin A can induce these tight junction proteins, protecting the barrier function. Additionally, vitamin A deficiency exacerbates intestinal dysbiosis, while supplementation promotes a healthy gut microbiome and alleviates intestinal disease symptoms [24, 25].

Carotenoids are natural antioxidants, including lycopene, α-carotene, β-carotene, and lutein, found in various fruits and vegetables. Lycopene activates antioxidant response elements (ARE), promoting the synthesis of cellular enzymes to eliminate reactive oxygen species. It also regulates inflammatory mediator signaling pathways and activates antioxidant gene expression to exert anti-inflammatory effects [26]. A study based on NHANES data found that higher dietary lycopene intake significantly reduced the risk of chronic constipation in men, while increased dietary α-carotene intake reduced the risk in women. These differences may be related to hormonal and metabolic differences between genders [27]. Notably, lycopene's antioxidant effect is dose-dependent; it acts as a potent antioxidant at low doses but becomes a pro-oxidant at high doses, potentially increasing oxidative stress. Therefore, it is crucial to control lycopene supplement doses to avoid adverse effects [28]. Ahn et al. [29] found that lutein and zeaxanthin reduce pro-inflammatory cytokines like IL-1β, IL-6, and IFN-γ and increase anti-inflammatory cytokines like IL-10, thereby inhibiting intestinal inflammation, alleviating constipation symptoms, and improving overall gut health. Zinc and selenium are essential trace elements for intestinal health and immune function, but there are few studies on their

relationship with constipation. Our study did not observe significant associations between individual zinc or selenium intake and constipation. Previous research suggests interactions between antioxidants, where lycopene combined with resveratrol and vitamin E significantly enhances its stability [30]. Vitamin C can regenerate the chromanoxyl radical of vitamin E, restoring its antioxidant activity [31]. Zinc and vitamin A synergistically maintain intestinal epithelial tissue stability, and animal studies indicate that vitamin E and selenium together can upregulate serum vitamin A levels [32, 33]. These synergistic effects among antioxidants significantly enhance the body's resistance to oxidative stress, aligning with our findings that CDAI, rather than individual antioxidants, is more significantly associated with reduced constipation risk.

Our study highlights the potential positive role of dietary antioxidants in reducing constipation risk. However, solely relying on antioxidant intake to improve constipation is not advisable. A diet rich in high-fiber foods and reduced intake of high-sugar, high-fat processed foods can improve various intestinal issues, including constipation [34]. Beyond dietary factors, physical activity, lifestyle habits, and psychological factors also significantly influence constipation, and managing these aspects can help improve intestinal health [35]. Primary care physicians should adopt a comprehensive approach when advising patients on maintaining gut health and preventing constipation. This approach should emphasize the importance of a diet rich in antioxidants, adequate hydration, regular physical activity, and effective stress management. By integrating these strategies, healthcare providers can offer a more effective and sustainable method for managing constipation and enhancing overall digestive health. Our study has several strengths. We utilized all available continuous NHANES data, providing a large, nationally representative sample, enhancing the reliability and generalizability of our results. The CDAI comprehensively considers multiple antioxidants, effectively reflecting overall dietary antioxidant intake and improving result accuracy. This study is the first to identify a significant association between CDAI and reduced constipation incidence, offering new insights into the potential role of dietary antioxidants in the pathophysiology of constipation and emphasizing the importance of dietary factors in gut health management.

However, our study also has limitations. As a cross-sectional study, it can only demonstrate an association between CDAI and reduced constipation incidence; further prospective studies or intervention trials are needed to establish causality. The data are based on self-reported information, which may be subject to recall or reporting biases. Dietary nutrient intake was averaged from two 24-hour dietary recalls, which may not accurately reflect long-term dietary habits. Finally, the results are primarily applicable to the U.S. population, and dietary habits, environmental factors, and socioeconomic conditions in different countries or regions may affect constipation prevalence, limiting the generalizability of our findings.

## 5 Conclusion

In conclusion, our study reveals a negative correlation between CDAI and the prevalence of constipation in a general adult population. This suggests that higher antioxidant intake may positively impact gut health and reduce constipation risk. The findings underscore the importance of antioxidants in a balanced diet. Future prospective and experimental studies from diverse regions and ethnic groups are necessary to validate this association and explore its potential underlying mechanisms.

## Supporting information

**S1 File.**
(ZIP)

## Acknowledgments

Our team sincerely appreciates all the staff and participants whose invaluable contributions have greatly enhanced the NHANES data collection.

## Author Contributions

**Conceptualization:** Shouxin Wei.

**Data curation:** Shouxin Wei, Sijia Yu.

**Formal analysis:** Sijia Yu.

**Investigation:** Shouxin Wei, Sijia Yu, Yingdong Jia.

**Methodology:** Shouxin Wei, Sijia Yu, Yingdong Jia.

**Software:** Shouxin Wei, Yunsheng Lan, Yingdong Jia.

**Supervision:** Yingdong Jia.

**Validation:** Yunsheng Lan.

**Visualization:** Yunsheng Lan.

**Writing – original draft:** Shouxin Wei.

**Writing – review & editing:** Shouxin Wei, Sijia Yu.

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
