## [Decision Letter · Decision Letter 0]

28 Aug 2024

PONE-D-24-33188Association between the composite dietary antioxidant index and constipation: evidence from NHANES 2005-2010PLOS ONE

Dear Dr. Wei,

Thank you for submitting your manuscript to PLOS ONE. After careful consideration, we feel that it has merit but does not fully meet PLOS ONE’s publication criteria as it currently stands. Therefore, we invite you to submit a revised version of the manuscript that addresses the points raised during the review process.

We look forward to receiving your revised manuscript.

Kind regards,

Aleksandra Klisic

Academic Editor

PLOS ONE

**Journal Requirements:**

Reviewers' comments:

Reviewer's Responses to Questions

**Comments to the Author**

1. Is the manuscript technically sound, and do the data support the conclusions?

Reviewer #1: Yes

Reviewer #2: Yes

2. Has the statistical analysis been performed appropriately and rigorously? 

Reviewer #1: Yes

Reviewer #2: Yes

3. Have the authors made all data underlying the findings in their manuscript fully available?

Reviewer #1: No

Reviewer #2: No

4. Is the manuscript presented in an intelligible fashion and written in standard English?

Reviewer #1: Yes

Reviewer #2: Yes

5. Review Comments to the Author

**Reviewer #1:** The manuscript titled "Association between the composite dietary antioxidant index and constipation: evidence from NHANES 2005-2010" is well-written with appropriate methodology and interesting findings. I have some comments for improvement:

1- Define abbreviations in their first use and make sure that abbreviated forms are being used after the definition.

2- Mention the clinical utility of your findings for a primary care physician to the discussion section.

3- In the conclusions, suggest future studies on this topic to overcome your limitations.

**Reviewer #2: **Wei et al. have performed a study on the association between composite dietary antioxidant index and constipation using NHANES data. The manuscript is well-written and the findings are interesting. These are my comments:

- A paragraph summarizing the clinical take-home message of this manuscript should be added to the discussion.

- The references prior to 2010 could be updated with those after 2010 since they provide more up-to-date findings.

- The first paragraph of the discussion should mention the main findings of the manuscript.

6. PLOS authors have the option to publish the peer review history of their article (what does this mean?). If published, this will include your full peer review and any attached files.

Reviewer #1: No

Reviewer #2: No

---

## [Author Response · Author response to Decision Letter 0]

1 Sep 2024

Response to Reviewers

Dear Editor and Reviewers,

We would like to thank you for the time and effort you have put into reviewing our manuscript, titled " Association between the composite dietary antioxidant index and constipation: evidence from NHANES 2005-2010." We appreciate the valuable feedback and suggestions provided, which have greatly helped us improve our work. We have carefully considered all comments and have made the necessary revisions to address each of them. We highlighted all the revisions in red colour.Below, we provide a detailed point-by-point response to each comment.

Reviewer #1:

1.Comment: Define abbreviations in their first use and make sure that abbreviated forms are being used after the definition.

Response: Thank you for this suggestion. We have amended the article to define abbreviations the first time they are used and to ensure that the abbreviated form is used after the definition. 

Changes in Manuscript: For example, we removed more than definitions in the introductory section, making sure to use the abbreviated form after the first definition

2.Comment: Mention the clinical utility of your findings for a primary care physician to the discussion section. 

Response: We agree with the reviewer's suggestion and have increased clinical relevance for PCPs in the discussion sections. This change enhances the manuscript's applicability to primary care settings by providing more targeted insights and practical recommendations that are directly relevant to primary care physicians.

Changes in Manuscript: We added clinical utility for the study's primary care physicians in the discussion section.

3.Comment: In the conclusions, suggest future studies on this topic to overcome your limitations.

Response: Thank you for bringing this to our attention. We have added future research directions to address our research shortcomings.

Changes in Manuscript: In the concluding section, we emphasise the importance of future research in different regions and populations.

Reviewer #2:

1.Comment: A paragraph summarizing the clinical take-home message of this manuscript should be added to the discussion.

Response: Thank you for this suggestion. In our article, we added a paragraph to discuss the clinical points of the study in terms of clinical prevention and treatment of constipation.

Changes in Manuscript: In the discussion section, we emphasised the importance of antioxidant diet, exercise and other factors in the prevention of constipation, and stressed that holistic health management is an important strategy for the prevention and relief of constipation.

2.Comment: The references prior to 2010 could be updated with those after 2010 since they provide more up-to-date findings.

Response: Thank you for highlighting this. We have made changes to the citations.The updated manuscript now reflects this improvement.

Changes in Manuscript: References prior to 2010 have been updated with more recent studies.

3.Comment: The first paragraph of the discussion should mention the main findings of the manuscript.

Response: We appreciate this suggestion and have revised the first paragraph of the discussion to ensure that the first paragraph of the discussion section refers to the main findings of the paper.

Changes in Manuscript: Revise the main findings of the study to the first paragraph so that the reader can see the results at a glance.

We believe these revisions have significantly strengthened the manuscript, and we hope that it now meets your expectations for publication. We are grateful for the constructive feedback and remain at your disposal for any further questions or clarifications.

Thank you once again for your consideration of our work.

Sincerely,

Shouxin Wei

Department of Gastrointestinal Surgery,Suining Central Hospital,Suining,China

---

## [Decision Letter · Decision Letter 1]

16 Sep 2024

Association between the composite dietary antioxidant index and constipation: evidence from NHANES 2005-2010

PONE-D-24-33188R1

Dear Dr. Wei,

We’re pleased to inform you that your manuscript has been judged scientifically suitable for publication and will be formally accepted for publication once it meets all outstanding technical requirements.

Kind regards,

Aleksandra Klisic

Academic Editor

PLOS ONE

Additional Editor Comments (optional):

Reviewers' comments:

Reviewer's Responses to Questions

**Comments to the Author**

1. If the authors have adequately addressed your comments raised in a previous round of review and you feel that this manuscript is now acceptable for publication, you may indicate that here to bypass the “Comments to the Author” section, enter your conflict of interest statement in the “Confidential to Editor” section, and submit your "Accept" recommendation.

Reviewer #1: All comments have been addressed

Reviewer #2: All comments have been addressed

2. Is the manuscript technically sound, and do the data support the conclusions?

Reviewer #1: (No Response)

Reviewer #2: (No Response)

3. Has the statistical analysis been performed appropriately and rigorously? 

Reviewer #1: (No Response)

Reviewer #2: (No Response)

4. Have the authors made all data underlying the findings in their manuscript fully available?

Reviewer #1: (No Response)

Reviewer #2: (No Response)

5. Is the manuscript presented in an intelligible fashion and written in standard English?

Reviewer #1: (No Response)

Reviewer #2: (No Response)

6. Review Comments to the Author

Reviewer #1: (No Response)

Reviewer #2: (No Response)

7. PLOS authors have the option to publish the peer review history of their article (what does this mean?). If published, this will include your full peer review and any attached files.

Reviewer #1: No

Reviewer #2: No

---

## [Editor Report · Acceptance letter]

19 Sep 2024

PONE-D-24-33188R1 

PLOS ONE

Dear Dr. Wei, 

I'm pleased to inform you that your manuscript has been deemed suitable for publication in PLOS ONE. Congratulations! Your manuscript is now being handed over to our production team.

Kind regards, 

on behalf of

Dr. Aleksandra Klisic 

Academic Editor

PLOS ONE